# Proteomic Analysis of Thin Filament Components Elucidates Changes in Spastic Muscle Sarcomere After Stroke

**DOI:** 10.3390/ijms262110356

**Published:** 2025-10-24

**Authors:** Yun He, Guangrun Liu, Junxi Wu, Xiaolin Jiang, Shengbo Yang

**Affiliations:** Department of Anatomy, Zunyi Medical University, 6 West University Road, Xinpu New Developing Area, Zunyi 563099, China; heyun03268767@163.com (Y.H.);

**Keywords:** stroke, spastic muscle, sarcomere, thin filament, proteomic analysis

## Abstract

This study analyzed the changes in the composition of thin filaments in spastic muscles after stroke to investigate the mechanism underlying changes in the sarcomeres. Twenty-four rats were randomly divided into four groups: normal, and 3, 6, and 9 days after stroke. A model of post-stroke gastrocnemius muscle spasm was created. Quantitative proteomic procedure and bioinformatics analysis revealed significant changes in cytoskeletal protein expressions in gastrocnemius muscles of each stroke group, particularly those on thin filaments. On the 3rd day after stroke, proteins upregulated within the thin filaments included actin-binding LIM protein 1, tropomyosin 3, leiomodin 2, drebrin-like protein, parvin beta, capping actin protein-gelsolin like, actinin alpha 2, and PDZ-LIM-domain protein 1, while downregulated proteins included tropomyosin 1, gelsolin, actinin alpha 3, and PDZ-LIM-domain protein 7. On the sixth day, upregulation of tropomyosin 2 was newly added while parvin alpha, destrin, PDZ-LIM-domain protein 3, leiomodin 3 were downregulated. On the 9th day, actinin alpha 2, PDZ-LIM-domain protein 7, and cofilin 2 were downregulated. These altered proteins are capable of promoting actin filament elongation and regulating Z-disc growth, and changes in their expression may be responsible for the changes in spastic muscle sarcomere after stroke.

## 1. Introduction

The global incidence of stroke has been reported to be 157.99/100,000 [1], and over 65% of patients develop secondary muscle spasm [2]. Research has shown that after stroke, the sarcomere length of spastic muscles increases, whereas the number and width of sarcomeres decrease [3]. An increase in sarcomere length was observed when the hind limbs of mice were subjected to simulated mechanical stress. However, the underlying mechanism by which these skeletal muscles undergo secondary sarcomeric changes in response to altered neural control remains poorly understood.

Thin filament length is an important determinant of the force–sarcomere length [4]. Actin filaments constitute the main stem of thin filaments, the length of which depends on the polymerization and dissociation of actin [5]. Thin filaments are fixed to the Z-disc, and some components of the thin filaments can affect Z-disc development. The growth and development of the long axis of the Z-disc affect the width of the sarcomeres, which determines the diameter of the muscle fibers [6].

The length of the actin filaments changes dynamically. Tropomodulin and leiomodin (Lmod) competitively bind to actin at the tips of actin filaments and regulate the dissociation and polymerization of actin, thereby controlling the length of thin filaments [7]. Tropomyosin, located on the lateral side of actin filaments, stabilizes their structure by preventing cofilin depolymerization [8]. PDZ-LIM-domain proteins involved in the composition of thin filaments affect the structural stability of thin filaments and the growth of Z-discs, thereby resulting in changes in muscle fiber diameter during different physiological functions [9].

Based on this, we hypothesized that the increase and decrease in the length and width, respectively, of spastic muscle sarcomeres after stroke may be caused by changes in protein expression within the thin filaments. We used a rat stroke model to build spastic gastrocnemius and used quantitative proteomics to analyze changes in thin filament composition. Our study is expected to provide a theoretical basis for analyzing the mechanisms underlying changes in spastic muscle sarcomeres after stroke and to guide future research on muscle spasms.

## 2. Results

### 2.1. Evaluation of Successful Establishment of Post-Stroke Muscle Spasm Model in Rats

#### 2.1.1. Gait

After the creation of the gait line plug model, the rats showed obvious limping, with the contralateral knee and plantar area in flexion states and the front and rear paws curled up. While walking, they tilted towards the contralateral side and performed circular movements.

#### 2.1.2. Zea-Longa Score

The normal group scored 0 points, while the groups on the 3rd, 6th, and 9th day after stroke scored (2.11 ± 0.60), (2.28 ± 0.67), and (2.15 ± 0.46) points, respectively. Compared to the normal group, the differences in the stroke groups were statistically significant with *p* < 0.05.

#### 2.1.3. Muscle Tone Detection

The changes in muscle tone in each group are shown in Figure 1. Compared to that in the control group, the tension in the gastrocnemius muscle in rats on days 3, 6, and 9 after stroke increased, especially on day 6, and the difference was statistically significant *(p* < 0.05). The difference in tension between the 6th and 9th day after stroke was not statistically significant (*p* > 0.05).

### 2.2. Changes in Sarcomere Length and Width After Stroke

Compared to that in the normal group, the sarcomere length increased and width decreased in the 3rd, 6th, and 9th day groups after stroke, with the 6th day group showing the most significant changes. The sarcomere length increased significantly by 3.16%, 7.89%, and 6.32%, while the sarcomere width narrowed by 24.81%, 46.85%, and 37.99%, respectively, in the 3rd, 6th, and 9th day groups after stroke (*p* < 0.05). The difference between the 6th day and 9th day after-stroke groups was not significant (*p* > 0.05) (Figure 2).

### 2.3. Proteomic Analysis

Gastrocnemius muscle samples were subjected to proteomic analysis, with a total of 20,830 peptide segments, 9373 specific peptide segments, 3050 proteins, and 3046 quantitatively comparable proteins. The peptides were mainly composed of 7–20 amino acids (Figure 3a), and most proteins corresponded to two or more peptide segments (Figure 3b). The coverage of most proteins was below 30% (Figure 3c), and the molecular weights of the proteins were distributed across different ranges (Figure 3d). Pearson correlation coefficients between all samples were calculated pairwise and a heatmap was plotted, as shown in Figure 3e.

### 2.4. Screening of Differentially Expressed Proteins

In this experiment, the number of differentially expressed proteins screened by predetermined fold changes on the 3rd, 6th, and 9th days after stroke was 406 (upregulated, 288; downregulated, 118), 530 (upregulated, 336; downregulated, 194), and 466 (upregulated, 232; downregulated, 234), respectively.

### 2.5. GO Annotation

The largest number of differentially expressed proteins were annotated as “intracellular anatomical structure,” “protein binding,” and “regulation of biological processes” in the categories of “cellular component,” “molecular function,” and “biological process,” respectively. On days 3, 6, and 9 post-stroke, the numbers of differentially expressed proteins in these categories were as follows: “intracellular anatomical structure,” 372, 477, and 412; “protein binding,” 244, 296, and 213; and “regulation of biological processes,” 275, 336, and 281 (Figure 4).

### 2.6. KOG Annotation

On the third day after stroke, 344 differentially expressed proteins were annotated using KOG. The Z class protein, annotated as the physiological function of the cytoskeleton, ranked second, accounting for 11.33% of the proteins (39/344). On the 6th and 9th days after stroke, the protein annotated as class Z ranked fourth, accounting for 8.33% (41/492) and 8.25% (33/400) of the proteins, respectively (Figure 5).

### 2.7. Subcellular Location Analysis

The identified Z proteins were annotated as thin filaments (ranked first), cytoplasm, Z-line, myosin complex, nucleus, microtubules, autophagosome membrane, and troponin complex. Three days post-stroke, proteins showing significant differences in expression on skeletal muscle thin filaments were annotated as follows: actin-binding LIM protein 1 (Ablim1), PDZ-LIM-domain protein 1 (Pdlim1), tropomyosin 3 (Tpm3), leiomodin 2 (Lmod2), drebrin-like protein (Dbnl), actinin alpha 2 (Actn2), parvin beta (Parvb), and capping actin protein-gelsolin like (Capg) and were upregulated. actinin alpha 3 (Actn3), tropomyosin 1 (Tpm1), and gelsolin (Gsn) were downregulated. Six days post-stroke, differentially expressed thin filament proteins were annotated as follows: Ablim1, Pdlim1, Lmod2, Dbnl, Parvb, Pdlim3, Tpm2, parvin alpha (Parva), and destrin (Dstn) were upregulated, while Lmod3 was downregulated. Nine days post-stroke, proteins annotated as thin filaments were upregulated, including Ablim1, Dbnl, Parvb, Capg, and Pdlim3; Actn2, Pdlim7, Tpm1, and cofilin 2 (Cfl2) were downregulated (Figure 6).

### 2.8. Protein–Protein Interaction Analysis

Based on the STRING database, the interaction relationships between the significantly differentially expressed proteins on post-stroke days 3, 6, and 9 and the differentially expressed thin filament proteins identified in this study were predicted. The results indicated that the differentially expressed thin filament proteins in each post-stroke group had predicted interactions with troponin and myosin. In particular, on post-stroke days 3 and 9, Cysteine and glycine-rich protein 3 (Csrp3) was included in the analysis due to its expression meeting the differential screening criteria, and its interaction with the differentially expressed thin filament proteins was predicted in both cases. However, on post-stroke day 6, Csrp3 expression did not meet the criteria and was therefore excluded from the differentially expressed protein set at this time point, resulting in no predicted interaction between Csrp3 and the differentially expressed thin filament proteins. By contrast, the proteins paxillin (Pxn), integrin alpha-7 (Itga7), and tensin 1 (Tln1) were predicted to interact with the differentially expressed thin filament proteins at this time point (Figure 7).

## 3. Discussion

Our understanding of the mechanism underlying post-stroke muscle spasms remains limited, and understanding the mechanisms underlying muscle spasms is necessary for developing muscle spasm treatments. Previous studies have shown that spastic muscles exhibit an increase in sarcomere length and decrease in sarcomere width after stroke [3]. Our results show that sarcomere changes are most pronounced 6 days after stroke in rats, with a 7.89% increase in sarcomere length and a 46.85% decrease in its width. The sarcomere is composed of regularly arranged thick and thin muscle filaments. Although the length of the thick filaments remains constant, the length of the thin filaments changes with the physiological function of the muscle fibers. Therefore, thin filament length is an important determinant of force–sarcomere length [10].

After successfully establishing a rat model of stroke with gastrocnemius muscle spasm, 3046 quantitatively comparable proteins were identified using quantitative proteomics, and their peptide length distribution, peptide quantity distribution, protein coverage distribution, and protein molecular weight distribution met quality control requirements. GO annotation analysis indicated that sarcomere changes after stroke are regulated by complex biological processes and are related to muscle sarcomere structural remodeling and protein–protein interactions. KOG annotation analysis revealed that differentially expressed proteins in each stroke group were concentrated within the cytoskeleton. Cellular component analysis further revealed that approximately one-third of the differentially expressed proteins in each stroke group belonged to the actin filament component.

From this, we speculated that the sarcomere changes occurring in spastic muscles after stroke are closely related to changes in thin filament components. Among these proteins, those related to changes in sarcomere length included the following. (1) Ablim1 anchors actin filaments to the Z-disc, thereby enhancing the stability of the sarcomere [11]. When Ablim1 expression increases, it promotes the polymerization of actin monomers, leading to the elongation of actin filaments and thin filaments. (2) Tpm helically wraps around actin filaments, thereby preventing Cfl2 from depolymerizing the actin filaments and stabilizing their structures [12]. Tpm3 expressed in slow-twitch fibers significantly inhibits Cfl2 by depolymerizing the actin filaments [13]. (3) Lmod2 is located at the tips of actin filaments and promotes ATP hydrolysis and provides energy for the aggregation of monomeric actin into actin filaments [14]. Actin filaments were prolonged when Lmod2 expression was upregulated [15]. (4) Lmod3 expression is downregulated after muscle spasms. The thin filaments in slow-twitch fibers are longer than those in fast-twitch fibers, thereby facilitating greater overlap between thin and thick filaments and resulting in a stronger contractile force in slow-twitch fibers [16]. However, whether Lmod3 promotes the transformation of fast-twitch fibers to slow-twitch fibers within spastic muscles and alters sarcomere length through the elongation of thin filaments [17] requires further investigation. (5) Dbnl can promote the aggregation of actin into actin filaments while maintaining the spatial relationship between actin and other proteins, such as alpha actin, tropomyosin, and cofilin, thereby enhancing the structural stability of actin filaments [18,19]. Therefore, the upregulation of Dbnl expression may be one contributor to actin filament elongation in spastic muscles, thereby leading to sarcomere elongation 3–9 days after stroke. (6) Parva and Parvb can bind alpha actin, integrin-linked kinase, and actin filaments and promote actin polymerization [20,21]. In the current study, Parva was upregulated on day 6 after stroke, whereas Parvb was upregulated on days 3, 6, and 9 after stroke. (7) Dstn can cleave actin filaments and restrict the aggregation of actin monomers, thereby regulating the length of actin filaments [22]. Six days post-stroke, Dstn expression was upregulated, actin filament elongation was inhibited, and sarcomere elongation began to decrease. This explains the recovery of sarcomere length 9 days after stroke. (8) Gsn and Capg are both located at the tips of actin filaments [23]. Gsn cleaves actin filaments [24,25], whereas Capg inhibits actin filament depolymerization [26]. Gsn expression was downregulated on the 3rd day after stroke, whereas Capg expression was upregulated on day 3 and 9; both changes promoted actin filament elongation.

The change in sarcomere width can be considered to indicate a change in muscle fiber diameter. Development of the Z-disc affects the diameter of muscle fibers [9]. In this study, two changes in the expression of proteins related to Z-disc formation were identified: Actn and Pdlim. Mammals have four subtypes of Actn: Actn1–Actn4. Among them, Actn2 and Actn3 exist in slow- and fast-twitch fibers of the skeletal muscle, respectively, near the Z-disc. They can bind to actin filaments through the N-terminal actin-binding domain, and the C-terminal calmodulin-like domain can bind to titin, stabilizing the spatial structure of actin filaments [27,28]. During skeletal muscle development, the growth of Z-discs depends on the involvement of Actn, and the thickness of Z-discs is directly proportional to the number of Actns anchored to the actin filaments [29,30]. In the current study, Actn2 expression was upregulated on the 3rd day after stroke, while Actn3 was downregulated. This change may be the result of the transition from fast- to slow-twitch fibers, which have a smaller diameter than fast-twitch fibers [31]; this explains the reduction in the sarcomere width of spastic muscles after stroke. The Pdlim family includes seven subtypes, Pdlim 1–7, of which Pdlim 1–4 belong to the actinin-associated LIM protein (ALP) subfamily and Pdlim 5–7 belong to the enigma homologous protein (Enigma) subfamily [32]. ALP family proteins inhibit Z-disc growth, whereas enigma family proteins promote their growth [9]. Moreover, Z-disc width determines sarcomere width. We showed that Pdlim 1 and Pdlim 3 were upregulated at all time points after stroke, whereas Pdlim 7 in the Enigma family was significantly downregulated nine days post-stroke. This suggests that the ALP and Enigma subfamilies, which inhibit and promote Z-disc growth, respectively, play important roles in narrowing spastic muscle sarcomere width after stroke.

Analysis of protein–protein interactions in thin filaments revealed interactions with various proteins in addition to the interactions with contraction proteins such as myosin and troponin. For example, Csrp3, a Z-disc protein, stabilizes actin filaments in the Z-disc and prevents Cfl2 depolymerization [33]. Csrp3 upregulation is involved in actin filament elongation, which stabilizes the connection between actin filaments and Z-discs. Pxn, Itga7, and Tln1 are upregulated after stroke. Pxn, Itga7, and Tln1 belong to the adhesive plaque complex, which connects the sarcomere and cell membrane through the laminin subunit α 2, thereby achieving bidirectional transmission of mechanical information between the sarcomere and extracellular matrix. This ensures that the internal structure of the sarcomere is not damaged by unbalanced forces [34].

Overall, the thin filament component proteins identified in this study promote actin filament elongation and regulate Z-disc growth. Changes in their expression may be required for the changes in spastic muscle sarcomere after a stroke. However, these findings are only preliminary. Although they lay the foundation for analyzing the molecular mechanisms underlying the changes in spastic muscle sarcomeres after stroke, further research elucidating these mechanisms is required.

## 4. Materials and Methods

### 4.1. Experimental Animals and Ethics

Based on the muscle spasm period from days 3 to 9 after stroke [35], specific pathogen-free healthy adult Sprague–Dawley (SD) rats were randomly divided into normal (n = 6) and stroke (n = 18) groups. The stroke group was further subdivided into 3-day, 6-day, and 9-day post-stroke groups (n = 6 each). All animal experimental protocols conformed to the guidelines for laboratory animal management (NIH publication No. 85–23) and were reviewed and approved by the animal care and use committee of Zunyi Medical University (approval number: ZMU21-2402-011, 21 February 2024). All methods were performed in accordance with the relevant guidelines and regulations.

### 4.2. Construction of Post-Stroke Muscle Spasm Model in SD Rats

The rats were anesthetized with 2–3% isoflurane. Following anatomical exposure of the common carotid artery and the proximal end of the external carotid artery, the proximal ends were ligated. A suture with a diameter of 0.26 mm (Beijing Jitai Yuancheng Technology Company, Beijing, China) was inserted to a depth of 18–20 mm, and the proximal end of the internal carotid artery was ligated with the suture. After model induction, rat limb movement was observed, and the Zea-Longa score was used to assess the extent of neurological damage. A BL-420N biological signal acquisition system (Chengdu Taimeng Software Company, Chengdu, China) was used to detect changes in muscle tone before and after the model establishment.

### 4.3. Mallory’s Phosphotungstic Acid-Hematoxylin Stain

The rats were deeply anesthetized via isoflurane inhalation and euthanized via cervical dislocation. To prevent interference with the length of the muscle fibers, the spasmodic gastrocnemius muscle, along with the knee and ankle joints, was removed. Immediately, a 20 mg muscle block was cut from the middle of the muscle belly for proteomic testing, and the remaining part was fixed using a 10% formaldehyde solution for 1 week. The gastrocnemius muscle was longitudinally cut into several pieces, softened in 25% sulfuric acid solution for 48 h, distilled, and washed with water for 5 min, separated into several single muscle fibers, soaked in 0.25% potassium permanganate solution for 2–3 min, distilled and washed with water for 1 min, soaked in 2% oxalic acid solution for 2–3 min, washed with water for 1 min, and stained with phosphotungstic acid-hematoxylin solution for 12 h. The samples were dehydrated using ethanol, made transparent with xylene, and sealed with neutral gum. Under an oil microscope, the length (length of the adjacent bright and dark bands) and width (width of the bright or dark bands) of 50 sarcomeres were continuously measured using the Motic Digilab II software Version 2.0 (manufactured by McDotti Industrial Group Company, Xiamen, China).

### 4.4. Protein Extraction and Digestion

The gastrocnemius muscle tissue was ground into a powder using a liquid nitrogen-precooled mortar. Four volumes of lysis buffer were added, followed by ultrasonic lysis. Sample extracts (six each) from the normal, 3-day, 6-day, and 9-day post-stroke groups were randomly and equally pooled into three replicates, labeled N-1 to N-3, D3-1 to D3-3, D6-1 to D6-3, and D9-1 to D9-3, respectively. Protein concentration was measured using the Bicin-Choninic Acid kit. Equal amounts of protein were used, and the volumes were adjusted consistently. Protein precipitation was performed using 20% trichloroacetic acid and incubated at 4 °C for 2 h, followed by centrifugation and removal of the supernatant. The precipitate was washed three times with acetone and air-dried. Tetraethylammonium bromide was added, followed by ultrasonic dispersion, trypsin digestion at a ratio of 1:50, and overnight enzymatic hydrolysis at 37 °C. Subsequently, dithiothreitol (reduction at 56 °C for 30 min) and iodoacetamide (incubation at room temperature in the dark for 15 min) were sequentially added for reduction.

### 4.5. Liquid Chromatography-Tandem Mass Spectrometry Analysis and Database Search

Peptide separation was performed using a NanoElute ultra-high performance liquid chromatography system (Bruker Corporation, Ettlingen, Germany). The mobile phases consisted of mobile phase A (0.1% formic acid in 2% acetonitrile) and mobile phase B (0.1% formic acid in 100% acetonitrile), with a gradient from 6% to 80% mobile phase B at a flow rate of 500 nL/min. After separation, the peptides were ionized via a capillary source and introduced into a timsTOF Pro 2 mass spectrometer (Bruker Corporation, Germany) for data acquisition. The Pulsar search engine in Spectronaut (v.17.0) was used to search against the *Rattus norvegicus* database. The false-positive rate of the proteins and peptides was strictly controlled at ≤1%. The criteria for screening differentially expressed proteins were: fold change >1.5 or <0.67 and *p*-value < 0.05.

### 4.6. Bioinformatics Analysis

Gene Ontology(GO) and Eukaryotic orthologous groups(KOG) annotations of differentially expressed proteins were performed using the EggNOG Database. Subcellular localization analysis was conducted via the Uniprot Database. The interaction network of significantly differentially expressed proteins was constructed using the STRING Database, with a confidence score threshold >0.7.

### 4.7. Statistical Processing

Statistical analysis of the experimental data was performed using the SPSS21.0 (IBM, Chicago, IL, USA) software. Data comparison between groups was conducted using one-way analysis of variance (ANOVA), with *p* < 0.05 indicating statistical significance.

## 5. Conclusions

Alterations in thin filament component proteins can promote actin filament elongation and regulate Z-disc growth, and changes in their expression may be responsible for changes in spastic muscle sarcomeres after stroke.

## Figures and Tables

**Figure 1 ijms-26-10356-f001:**
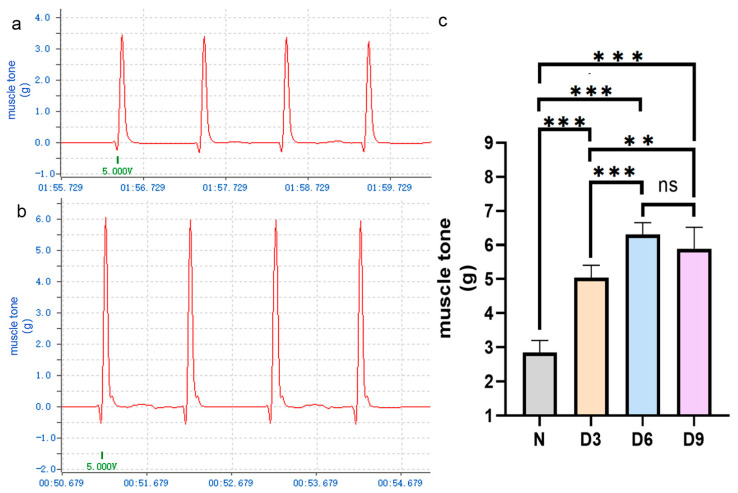
Comparison of gastrocnemius muscle tension among different groups of rats.(**a**,**b**) Representative curves of gastrocnemius muscle tension in the normal and stroke groups on day 6 post-stroke. (**c**) Comparison of gastrocnemius muscle tension among the different groups of rats. *n* = 6 per group. Error bars represent standard deviation. ** represents *p* ˂ 0.01, *** represents *p* ˂ 0.001, and ns represents *p* > 0.05.

**Figure 2 ijms-26-10356-f002:**
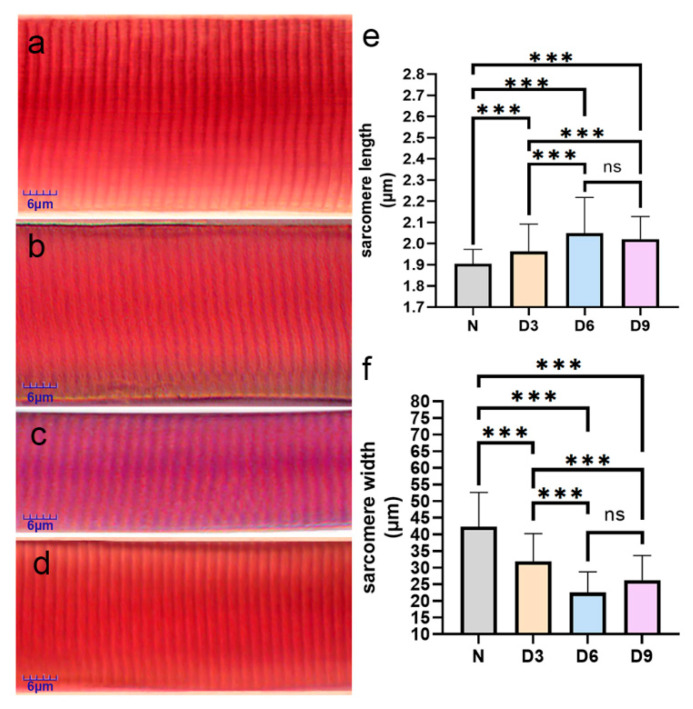
Comparison of sarcomeres changes from various groups. (**a**–**d**) Sarcomeres from the normal group, 3rd day post-stroke, 6th day post-stroke, and 9th day post-stroke, respectively, after Mallory’s phosphotungstic acid-hematoxylin staining. (**e**,**f**) Comparison of the changes in the length and width of muscle fibers in the gastrocnemius muscle of rats from each group. *n* = 6 per group. Error bars represent standard deviation. *** represents *p* ˂ 0.001 and ns represents *p* > 0.05.

**Figure 3 ijms-26-10356-f003:**
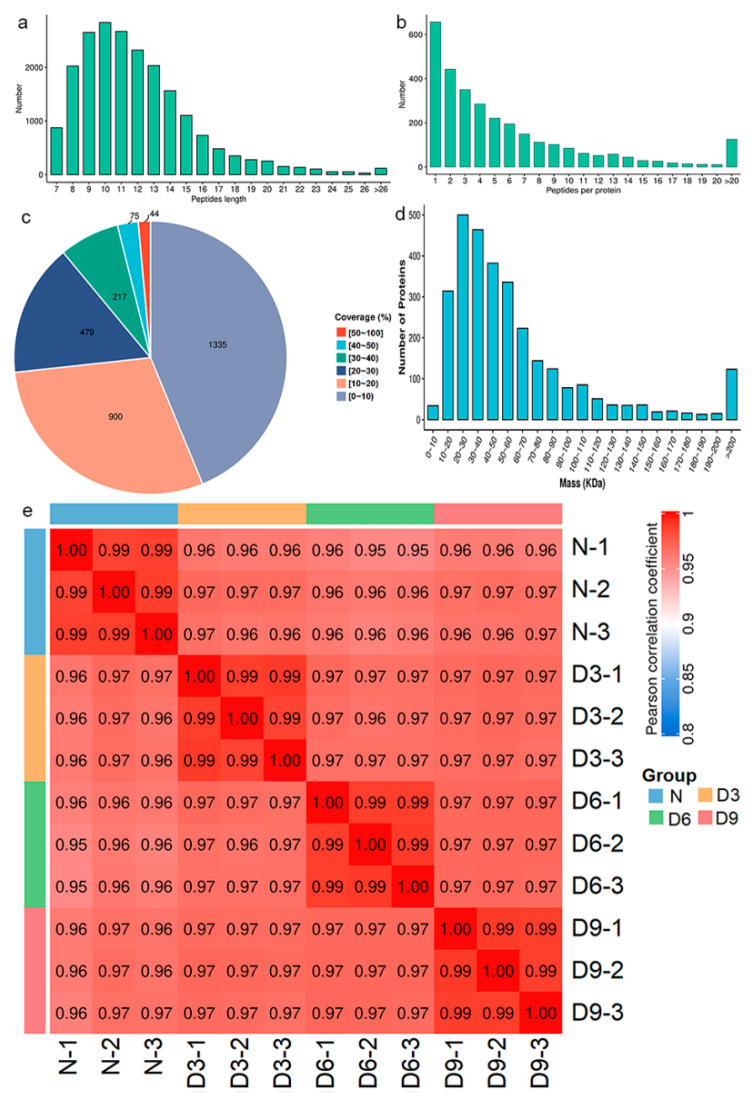
Protein identification and analysis. (**a**) Distribution of the peptide segment lengths. (**b**) Distribution of peptide segments. (**c**) Distribution of protein coverage. (**d**) Molecular weight distribution of the protein. (**e**) Pearson’s correlation coefficient heat map of samples.

**Figure 4 ijms-26-10356-f004:**
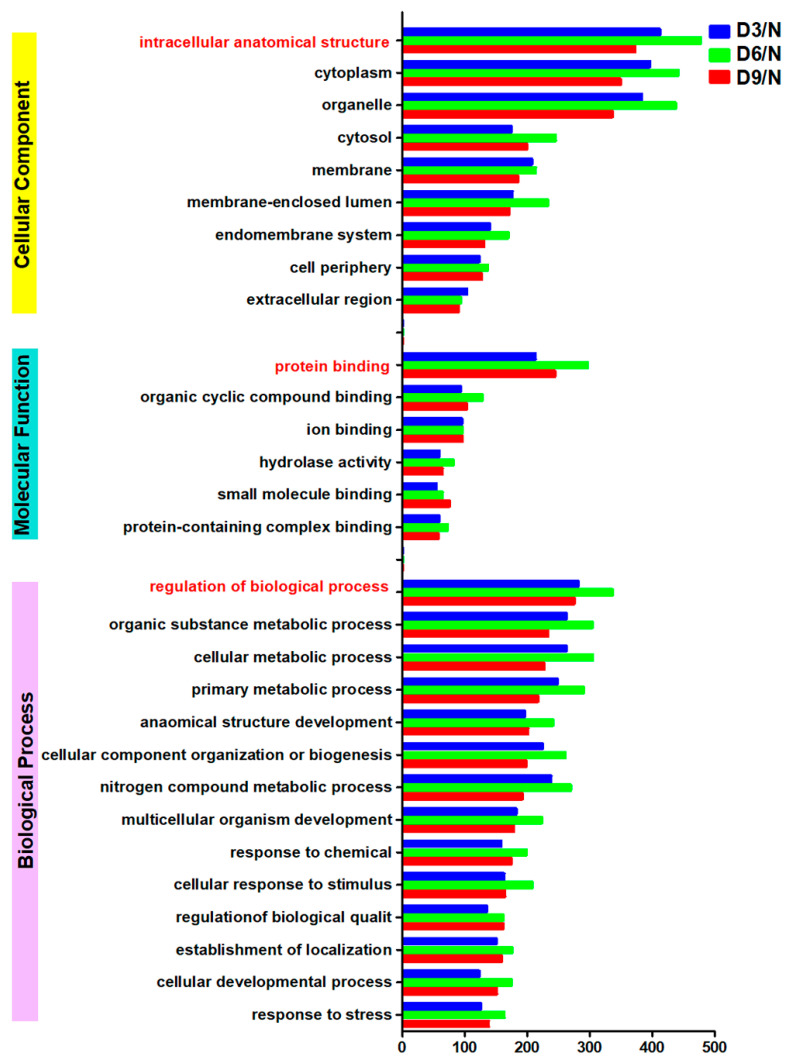
GO annotation.

**Figure 5 ijms-26-10356-f005:**
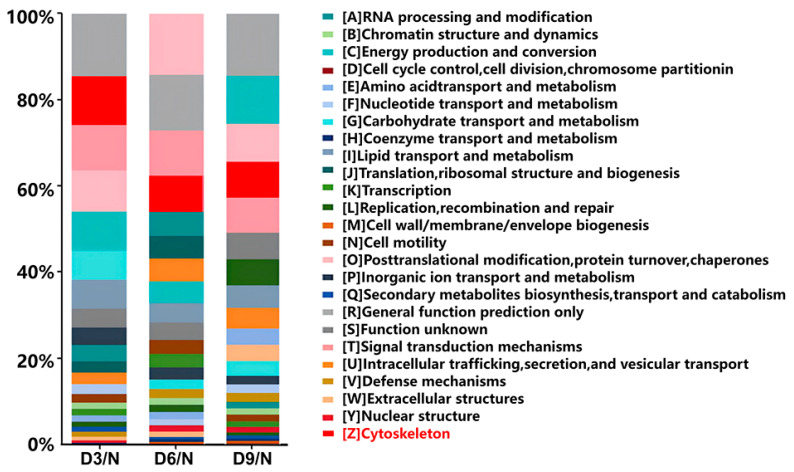
KOG annotation.

**Figure 6 ijms-26-10356-f006:**
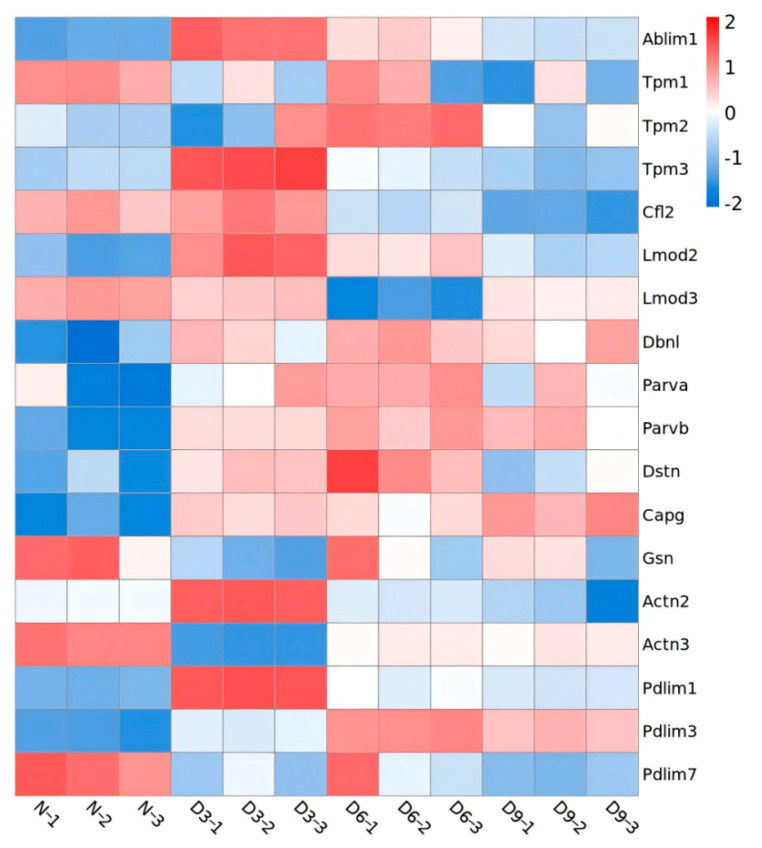
Heatmap of differential protein expression in thin filaments.

**Figure 7 ijms-26-10356-f007:**
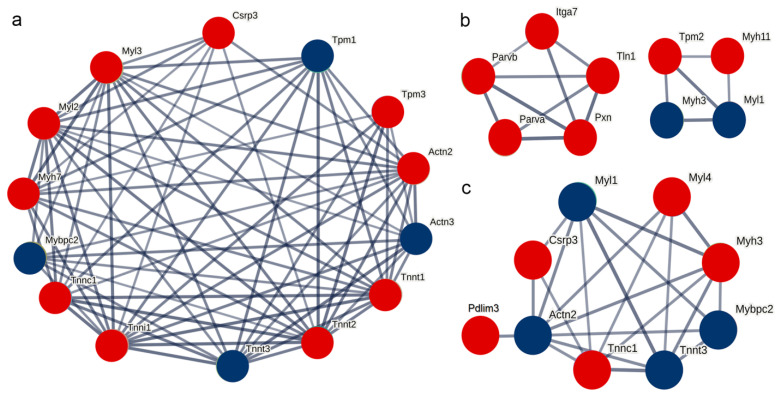
Protein–protein interaction analysis. (**a**–**c**) Interaction network analyses of the actin filament components on the 3rd, 6th, and 9th days post-stroke, respectively. Red and blue circles represent upregulated and downregulated expression, respectively.

## Data Availability

The datasets generated and analyzed during the current study are available from the corresponding author on reasonable request.

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
