# Peer review of "Proteomic Analysis of Thin Filament Components Elucidates Changes in Spastic Muscle Sarcomere After Stroke"

_ijms, 2025, doi:10.3390/ijms262110356_

Round 1
Reviewer 1 Report
Comments and Suggestions for Authors
The manuscript by Y. He et al. is the research on the animal model of muscle tissue affected by stroke. The authors performed the quantitative proteomic profiling of “spastic” muscle samples against healthy controls and identified a number of muscle proteins whose abundance was altered in the after-stroke model. These findings were analyzed and implied to explain macroscopic after-stroke muscle changes reported earlier.
In general, the work is solid and contains large experiment; however, I have some critical comments to the writing and presentation of the material.
- The problem that motivated this study is not sufficiently introduced. The authors use the terms “spastic muscles” or “gastrocnemius muscle spasm”, while in fact this study characterizes not the immediate result of a stroke-affected muscle but the dynamics of its adaptation to neurological defect. The changes in protein profiles reflect the sarcomere recovery from the stroke rather than the consequence of the stroke. This is unclear from the text in the current variant. The authors might consider a brief elaboration on this topic in the Introduction and Discussion.
- The results ( Figure 6 ) show that there are proteins whose expression level differs in the biological replicates. E.g., protein Tpm2 is significantly upregulated in one replicate and significantly downregulated in two other replicates; same observation is valid for the protein Gsn. First, these results are ignored in the text; the authors should analyze possible reasons of this discrepancy since it makes other results look less reliable. Second, probably such proteins should be grouped separately and not be included in the analysis together with the rest of proteins.
- The coverage of some protein sequences in the proteomic analysis is shown in Figure 3. It would be helpful to understand what coverage was obtained for the proteins with differential expression analyzed further in the text. Some proteins discussed by the authors (such as e.g. Pdlim1,3,7 proteins) have circa 50% sequence identity; at the coverage of <10% and peptide length of 10-11 residues, can the author confirm that they identified this particular protein and not its homolog?
- “Pearson correlation coefficients between all samples were calculated pairwise” (lines 109-110). Correlation coefficients between which exactly parameters of the samples?
- Protein Ablim7 is mentioned in the Results (line 148) but not included in the Discussion or Figure 6. Is there any reason for that?
- Interaction analysis is poorly described. (a) “…interactions with cysteine and glycine-rich protein 3 (Csrp3) were detected on the Z-disc” (lines 159-161). Interactions were not detected nor observed, since this analysis was computational, not experimental. (b) It is not quite clear how the interaction change from Day X to Day Y were found. (c) Interaction between which “components of thin filaments” were analyzed? The corresponding part of Materials and Methods does not allow to understand this analysis.
- First paragraph of the Introduction should be removed, as well as the phrase “The text continues here” (line 82).
- The text contains a number of incorrect phrases; e.g. “We constructed a rat stroke model using the spastic gastrocnemius muscle” (lines 59-60), “Gastrocnemius muscle samples were identified using proteomics” (line 104), “…the molecular weights of the proteins were distributed at different stages” (lines 108-109) etc. that should be corrected.
Overall, the manuscript can be recommended for publication after revision.
Author Response
Thank you all for your valuable feedback! Due to the character limit of the reply box, I have compiled all the responses into a PDF file and sent it to the reviewers for their reading

Reviewer 2 Report
Comments and Suggestions for Authors
Summary:
This paper presents novel and potentially important findings regarding the molecular mechanism of the changes in spastic muscle sarcomere after stroke. Findings from quantitative proteomic analysis concerning changes in protein levels of thin filament-associated proteins in the gastrocnemius muscle after stroke suggest novel mechanisms underlying changes in spastic muscle sarcomeres. This content is sufficient for publication in International Journal of Molecular Sciences, but minor revisions will be recommended.
Major Comments
- If possible, brain tissue images demonstrating successful cerebral infarction (2,3,5-triphenyl tetrazolium chloride staining or Nissl staining) would be beneficial.
- If possible, it would be helpful to show macroscopic images of the gastrocnemius muscle from each group.
Minor Comments
- Although the “Materials and Methods” section states that the n value for each group is 6, the figure legends for Figures 1 and 2 must include the n value. Also, are the error bars on the graphs standard errors or standard deviations?
- I believe the title's wording is too strong. The phrase “the mechanism underlying” should be removed. This is because no evidence has been presented here that the protein changes directly lead to changes in spastic muscle sarcomere.
Author Response

(The authors gave the same response as above.)
